# The Human-Restricted Isoform of the α7 nAChR, CHRFAM7A: A Double-Edged Sword in Neurological and Inflammatory Disorders

**DOI:** 10.3390/ijms23073463

**Published:** 2022-03-22

**Authors:** Simona Di Lascio, Diego Fornasari, Roberta Benfante

**Affiliations:** 1Department of Medical Biotechnology and Translational Medicine (BIOMETRA), Università degli Studi di Milano, 20129 Milan, Italy; simona.dilascio@unimi.it (S.D.L.); diego.fornasari@unimi.it (D.F.); 2CNR Institute of Neuroscience, 20845 Vedano al Lambro, Italy; 3NeuroMi, Milan Center for Neuroscience, University of Milano Bicocca, 20126 Milan, Italy

**Keywords:** CHRFAM7A, CHRNA7, human-specific gene, nicotinic receptor, cholinergic anti-inflammatory pathway, neurodegeneration, neuroinflammation

## Abstract

*CHRFAM7A* is a relatively recent and exclusively human gene arising from the partial duplication of exons 5 to 10 of the α7 neuronal nicotinic acetylcholine receptor subunit (α7 nAChR) encoding gene, *CHRNA7*. *CHRNA7* is related to several disorders that involve cognitive deficits, including neuropsychiatric, neurodegenerative, and inflammatory disorders. In extra-neuronal tissues, α7nAChR plays an important role in proliferation, differentiation, migration, adhesion, cell contact, apoptosis, angiogenesis, and tumor progression, as well as in the modulation of the inflammatory response through the “cholinergic anti-inflammatory pathway”. *CHRFAM7A* translates the dupα7 protein in a multitude of cell lines and heterologous systems, while maintaining processing and trafficking that are very similar to the full-length form. It does not form functional ion channel receptors alone. In the presence of *CHRNA7* gene products, dupα7 can assemble and form heteromeric receptors that, in order to be functional, should include at least two α7 subunits to form the agonist binding site. When incorporated into the receptor, in vitro and in vivo data showed that dupα7 negatively modulated α7 activity, probably due to a reduction in the number of ACh binding sites. Very recent data in the literature report that the presence of the duplicated gene may be responsible for the translational gap in several human diseases. Here, we will review the studies that have been conducted on *CHRFAM7A* in different pathologies, with the intent of providing evidence regarding when and how the expression of this duplicated gene may be beneficial or detrimental in the pathogenesis, and eventually in the therapeutic response, to *CHRNA7*-related neurological and non-neurological diseases.

## 1. Introduction

*CHRFAM7A* (*dupα7*) is a human-lineage-specific gene [1,2,3] that first appeared about 3.5 million years ago during evolution: this can be confirmed by its absence in other primates [4], and its absence in rodents excludes the possibility of gene loss [5]. It arose from the duplication of *CHRNA7* gene exons 5 to 10 (Figure 1A), encoding the α7 subunit of the neuronal nicotinic acetylcholine receptor (nAChR) on chromosome 15q13-q14, 1.6 Mb apart from *CHRNA7,* and has an inverted orientation with respect to the parental gene (Figure 1B) [2,6,7]. Here, the *CHRNA7*-derived cassette fused in-frame to a cluster (*FAM7A*) of seven exons (A to F), which are present four times on chromosome 15q13.3, located both upstream and downstream of *CHRNA7*, in the sense and antisense directions (Figure 1A). In particular, exons A, B, C, and E derive from the unc-51 like kinase 4 gene (*ULK4*), located at chromosome 3p22.1, whereas exons D and F are homologous to the *GOLGA8B* gene, which is located 2.5 Mb 3′ from *CHRNA7* [8].

More recently, a polymorphism (c.497-498delTG; rs67158670) and a deletion of two base pairs (*Δ2bp)* in exon 6, occurred in *CHRFAM7A* (dupΔα7; Figure 1C), which was determined as being associated with gene inversion, with the same orientation as the *CHRNA7* gene [6]. The *CHRFAM7A* gene varies in terms of its number of copies in each individual, with 5–10% of individuals being non-carriers of *CHRNA7* duplication and approximately 30% only having one copy [2,9,10]. The absence, as well as the presence of one or two copies of the gene, and the presence of *Δ2bp* deletion, results in six different genotypes (see Box 1). Moreover, the frequency of the *Δ2bp* allele varies in different ethnic groups (Table 1), and this polymorphism has been associated with an increased susceptibility to psychiatric disorders (see below).

Box 1The six possible combinations of the *CHRFAM7A* genotype.• *CHRFAM7A null/null*• *CHRFAM7A dupα7/null*• *CHRFAM7A dupα7/dupα7*• *CHRFAM7A dupα7/dupΔα7*• *CHRFAM7A dupΔα7/dupΔα7*• *CHRFAM7A dupΔα7/null*

Recently, data in the literature have reported that the presence of the duplicated gene may be responsible for the translational gap in several human diseases [11,12,13,14]. However, it is important to address whether this isoform is indeed expressed in vivo. Moreover, clinical and pre-clinical data neither seem to indicate a unique role of *CHRFAM7A,* nor whether it is protective or toxic, because of confounding factors caused by the disease being studied, and the model (clinical or pre-clinical) used. In this review, we will summarize all of the biochemical studies containing evidence of the formation of a heteromeric receptor composed of *CHRNA7* and *CHRFAM7A* encoded subunits for the first time. Moreover, we present all of the available data on the role of *CHRFAM7A* in different pathologies, with the intent of providing evidence regarding when and how the expression of this duplicated gene may be beneficial or detrimental in the pathogenesis, and eventually in the therapeutic response, to both neurological and non-neurological diseases. As a common trait, the deregulated expression of both genes (up- and down-regulated), due to the presence of the *Δ2bp* polymorphism in *CHRFAM7A* and/or the CNV of both genes, alter the *CHRNA7/CHRFAM7A* ratio, thus contributing to the beneficial/detrimental effects of dupα7 on α7 nAChR function.

## 2. Structure of Neuronal Nicotinic Receptors (nAChRs)

Nicotinic Acetylcholine Receptors (nAChRs) are part of the *Cys-loop* receptor superfamily. These receptors are transmembrane ion channels activated by neurotransmitters that are mainly expressed in the central and peripheral nervous systems and are responsible for the regulation and transmission of excitatory and inhibitory signals [15]. The ionotropic GABAergic receptors GABA_A_ and GABA_C_, glycinergic receptors, serotonin (5-HTn) receptors, and the aforementioned nAChRs, which include muscle- and neuronal- subtypes, are the best-known members of the *Cys-loop* receptor superfamily [16]. *Cys-loop* receptors share a common pentameric structure, with their subunits being arranged in a pseudo-symmetrical rosette, forming a central pore [15,16]. Each subunit consists of an extracellular hydrophilic N-terminal domain that contains the binding site for the ligand, three hydrophobic transmembrane domains (M1–M3), a highly variable cytoplasmic domain between the different subunits, a fourth hydrophobic domain (M4), and an extracellular C-terminal domain (Figure 2). The M2 domains of the five subunits form the central pore and possess amino acids that are important for ion selectivity, permeability, and channel gating [16]. Nicotinic receptors are selective cationic channels for Na^+^, K^+^, and Ca^2+^, and have a high affinity for nicotine and the endogenous ligands choline (Ch) and acetylcholine (ACh); the relative affinity of the ligands varies according to the subunit composition [16]. The combination of nine α (α2–α10) and three β (β2–β4) subunits form both hetero- and homo-pentamer receptors that have different structural, functional, and pharmacological properties (Figure 2). Homo-pentameric receptors have five identical orthosteric ACh binding sites that are located in the extracellular interfaces between two adjacent subunits (Figure 2), whereas hetero-pentameric receptors, which have two α/three β and three α/two β subunits, have two orthosteric binding sites that are located at the interfaces between the α subunit and the β subunit (Figure 2).

nAChRs, as modulators of the release of neurotransmitters, play a fundamental role in cognitive functions, and their decline or dysfunction leads to neuropsychiatric and neurodegenerative diseases. Genetic and knock-out studies on mouse models have confirmed a link between the malfunction or mutation of these genes and pathologies such as epilepsy, schizophrenia, depression, and nicotine addiction [16,17].

### 2.1. The α7 nAChR

α7 nAChR is a homo-pentamer whose subunit, α7, is encoded by the ten-exon *CHRNA7* gene (Chr 15q13-q14), which allows the translation of a protein of about 57 kDa (Figure 3A). Exons 1–7 encode the N-terminal portion that contains the ACh binding site; exons 7–8 comprise the M1–M3 transmembrane regions; and exons 9–10 comprise the cytoplasmic loop, the fourth transmembrane domain (M4), and the C-terminal portion (Figure 3A). α7 nAChR exhibits a low affinity for nicotine, a high affinity for α-bungarotoxin (α-Bgtx), and is highly permeable to Ca^2+^, thus suggesting that it plays a role not only as an ion channel, but also as a regulator of the calcium-activated signaling pathways [16]. The receptor is expressed in the brain, in excitatory and inhibitory neurons, both pre-synaptically, where it modulates the release of neurotransmitters, and post-synaptically, as well as in astrocytes and microglia [18,19,20]. In the periphery, it is expressed in neuroendocrine cells [21], sperm acrosome [22], epithelial cells [23], microvascular endothelial cells [24], pulmonary fibroblasts [25], bone marrow [26], chondrocytes [27], T and B lymphocytes [28,29], and macrophages [30]. Here, it plays an essential role in the “cholinergic anti-inflammatory” response [31,32,33]. Its nature as a modulator of intracellular calcium concentration, which is one of the most widespread second messengers, explains its almost ubiquitous expression. *CHRNA7* is related to several disorders with cognitive deficits, including schizophrenia [9], P50 auditory gating deficits [34], autism [35], epilepsy [36], bipolar disorder [37], attention deficit hyperactivity disorder (ADHD) [38], Down syndrome [39], Parkinson’s disease (PD) [40], and Alzheimer’s disease (AD) [41,42]. In extra-neuronal tissues, α7 nAChR plays important roles in proliferation, differentiation, migration, adhesion, cell contact, apoptosis, angiogenesis, and tumor progression [43,44,45,46,47,48].

### 2.2. The dupα7 Receptor

The *CHRFAM7A* gene transcribes for two isoforms (isoform 1, NM_139320.2/NP_647536.1; isoform 2, NM_148911.1/NP_683709.1) that differ in terms of the presence of exon B [49], due to alternative splicing. The transcript of isoform 1 presents two open reading frames (ORF) (Figure 3B); translation starting from exon B (ORF 1) results in a protein that is 412 amino acids (aa) long (46 KDa; dupα7) and is different from the α7 nAChR subunit for the N-terminal domain which is formed by a 27 aa long peptide (Figure 3B), but it still includes the disulfide bridge and the vicinal cysteines. The resulting peptide will lose a putative glycosylation site and part of the agonist binding site. A shorter polypeptide (38 KDa) may putatively originate from the second ORF starting at exon 6 (Figure 3B). This translation start site is the only start site that exists in isoform 2 (Figure 3C) and results in the 38 KDa protein, which lacks the signal peptide and the entire binding site, but contains all of the α7 transmembrane domain sequences. The expression of both transcripts and proteins has been confirmed in both heterologous expression systems [50,51,52] and in vivo [49,50,53,54,55,56]. The *Δ2bp* deletion polymorphism (Figure 3D) introduces a stop codon in ORF 1, generating a truncated subunit that is formed by a short 40 aa long peptide that is codified by exon B–exon 5. It is possible to synthesize the 38 KDa protein (dupΔα7) if translation restarts from exon 6, downstream from the *Δ2bp* polymorphism (Figure 3D) [51,52].

In leukocytes, a ~2 Kb long promoter, located upstream and that overlaps with part of exon D, drives *CHRFAM7A* expression [49].

Numerous attempts have been made to understand the nature and function of this duplicated gene. The homology of the full-length gene suggests that the duplicated gene may have a new function, but always in the context of cholinergic biology. The fact that this endogenous duplicated gene translates a subunit [49,53,57,58], reinforces the hypothesis that dupα7 may interact with and regulate α7 nAChR.

### 2.3. The Heteromeric α7/dupα7 Receptor

*CHRFAM7A* gene products (mRNA and protein) are present in different brain areas in different proportions compared to *CHRNA7* (up to 10–20% of the α7 transcript) [1,8,44,50,55,59,60,61], as well as in at least thirty tissues in the periphery, including in leukocytes [11,45,49,53,54,56,57,62]. In leukocytes, *CHRFAM7A* expression overcomes that of *CHRNA7* by two orders of magnitude [49,50], and seminal electrophysiological studies [57] have failed to measure an ACh or a nicotine-evoked current, probably due to the missing ACh binding sites. On the other hand, the presence of the transmembrane domains in the translated dupα7 subunits (in particular M2, forming the ion pore) indicates that a hypothetical dupα7 receptor could trigger Ca^2+^ influx in response to a specific ligand. An alternative hypothesis is that the dupα7 subunit does not form homomeric receptors, but is instead assembled with conventional α7 subunits to form a new class of heteromeric receptors. The role of dupα7 in these receptors may include the formation of the pore as well as many other regulatory functions.

To address this issue, several groups have used both functional and structural approaches [50,51,52,58,63,64], most of which have been based on the heterologous expression of the α7 and dupα7 proteins in oocytes and cell lines. In oocytes, the expression of dupα7 alone has failed to produce any ACh-induced currents [50], an effect that has also been observed in mammalian cell lines [51,52,63], and that has been hypothesized as being due to the missing agonist binding sites and the lack of β1β2 loop, both of which are required for channel opening [65,66]. When the receptors were co-expressed, the increased concentration of dupα7 (α7/dupα7 ratio 1:5/1:10) reduced the nicotine- and ACh-induced α7 currents, but not receptor sensitivity to ACh [50,51,52], thus suggesting that dupα7 has a dominant negative effect on the α7 current. The overexpression of the construct with the *Δ2bp* in exon 6, dupΔα7, further reduced the current amplitude by an additional 10% [51]. This effect has not been replicated in the Neuro2a cell line, which may be due to poor dupα7 translation [52], something that was recently confirmed in SH-SY5Y cells [58].

Confocal images using both an antibody, which detects both receptors, and FITC conjugated α-Bungarotoxin (α-Bgtx), an α7 antagonist that only binds to the N-terminal domain of the α7 receptor, demonstrated that this effect was due to a reduced number of α7 receptors at the plasma membrane [11,50,63,64]. The dupα7 protein seemed to preferentially localize within the endoplasmic reticulum (ER) [50,52,64], making it likely that the dominant negative effect may be exerted by increased α7 receptor retention in the ER (Figure 4). This model does not exclude the possibility of a heteromeric receptor α7/dupα7 being formed. In the presence of ^125^I-α-Bgtx, surface receptor binding assays showed a reduced number of ^125^I-α-Bgtx binding sites and an increase in the equilibrium dissociation constant (*Kd)*, thus suggesting the formation of a less functional receptor [50,51]. On the other hand, higher *CHRFAM7A* expression versus *CHRNA7* expression can result in a potentiation of the effect of a positive allosteric modulator that is able to bind to the transmembrane domains of the receptor (PNU-120596) [51]. This finding proved that a heteromeric receptor that incorporates CHRFAM7A subunits, and with properties that are different from those of α7 nAChR, can be formed, although this result has not been confirmed by other groups [50,58,63].

The putative assembly of the subunits was investigated using *Forster Resonance Energy Transfer* (FRET) *by Donor Recovery after Acceptor Photobleaching* (DRAP), and *FRET by Fluorescence Lifetime Imaging Microscopy* (FLIM), in the Neuro2a [52], GH4C1 and RAW264.7 [64], and SH-SY5Y [58] cell lines. Both approaches revealed and confirmed the assembly of duplicated subunits with full-length α7, forming several types of hetero-pentamers between α7 and dupα7/dupΔα7 (α7 with α7; α7 with dupα7; α7 with dupΔα7; dupα7 with dupα7; dupΔα7 with dupΔα7). According to these data, a limited number of dupα7 subunits (mainly one) may be incorporated and are likely adjacent to a α7 subunit in the heteromeric receptor. Interestingly, the duplicated isoforms can interact with other nAChRs subunits, namely α4 and α3 [52].

The formation of α7/dupα7 or α7/dupΔα7 hetero-pentamers in cell lines does not imply that they are functional. This question has been addressed by means of the substituted cysteine accessibility method (SCAM), a technique that allows the measurement of altered ion channel function, following the substitution of key amino acids with cysteine, and treatment with sulfhydryl-specific reagents [52]. The leucine at the 247 -position resides nine residues downstream from the cytoplasmic beginning of the TM2 (9′ position), which forms the ion channel pore of α7 and the duplicated isoform. A whole-patch clamp of Neuro2a cells, transfected with α7 along with substituted L9′C subunits (α7 L9′C; dupα7L9′C; dupΔα7L9′C), showed a reduction in the ACh-evoked inward current (I_ACh_) in the presence of the dupα7 subunit isoforms that was similar to the reduction measured on cells expressing α7/α7L9′C. These data suggest that both dupα7 and dupΔα7 could form heteromeric functional receptors with α7. Despite the expression of dupα7 and dupΔα7, there was no effect on the ACh-evoked current amplitude in Neuro2a [52]. The inclusion of dupα7 induced a higher sensitivity to a high concentration of choline compared to cells that were only expressing α7 or α7/dupΔα7, whereas brief exposure to a physiological concentration of choline led to a faster desensitization of both the α7/dupα7 and α7/dupΔα7 receptors. Conversely, the inclusion of dupΔα7 increased the sensitivity of the α7 nAChR to the full α7 agonist varenicline [52], which is commonly used for smoking cessation [67]. However, genetic studies did not reveal any significant association between dupΔα7 and smoking abstinence [67].

Patch clamp analyses to measure single-channel function, coupled with the electric fingerprinting strategy, have provided the first evidence of the actual formation of functional hetero-pentamers [63]. This strategy relies on the possibility of distinguishing the activity of the α7 subunit with three arginine substitutions at the intracellular M3–M4 loop region (α7LC, α7 low-conductance). α7LC incorporation generates multiple and discrete current amplitude clusters, each one corresponding to a different population of receptors with a fixed number of LC subunits. LC receptors, when expressed individually, do not display detectable currents at the single-channel level. Single channel currents activated by ACh and 1 μM PNU-1205696, from cells expressing α7/α7LC, distribute into clusters of different amplitudes, something that is indicative of the formation of pentameric receptors that incorporate from zero up to three α7LC subunits. When co-expressed, α7LC + dupα7, or the reverse combination dupα7LC + α7, reveal that dupα7 can assemble with α7, forming functional heteromeric receptors containing one, two, or three dupα7 subunits (Figure 4). Mutagenesis analyses of the conserved Tyr-93 in the α7 subunit, which is important for ACh binding, in the context of the (α7)_2_(dupα7)_3_ receptor, revealed that the heteromeric receptor has to contain at least two adjacent α7 subunits, and no more than three dupα7 subunits to be functional (Figure 4) [63,68]. Moreover, the channel kinetics elicited by the ACh + PNU-120596 of the (α7)_2_(dupα7)_3_ receptor does not differ from that of the α7 homomeric receptor [63], as previously reported [51].

These findings have been confirmed by computational model studies [69] on all of the possible combinations of α7 and dupα7 pentamers. The authors of this study have shown that receptors with dupα7/α7 interfaces are more stable and less detrimental for ion conductance than dupα7/dupα7 interfaces, and that the most stable stoichiometry is 1: dupα7/4: α7; however, functional heteromeric receptors can be formed with no more than three dupα7 subunits (Figure 4). Moreover, the heteromeric receptors showed a very low affinity for α-Bgtx, thus explaining the reduction observed in previous studies [50,51,52,70], and are less sensitive to Aβ_1–42_, which supports recent experimental data [12].

In conclusion, these studies demonstrated that the duplicated form of the *CHRNA7* gene, *CHRFAM7A*, translates for the dupα7 (or dupΔα7) protein in a multitude of cell lines and heterologous systems with processing and trafficking that are very similar to the full-length form without affecting *CHRNA7* transcription. It does not form functional ion channel receptors alone. In the presence of *CHRNA7* gene products, dupα7 can assemble and form heteromeric receptors that, in order to be functional, should include at least two α7 subunits to form the agonist binding site. When incorporated into the receptor, dupα7 negatively modulates α7 activity, as demonstrated by a net reduction in the macroscopic current, which is inversely proportional to the amount of dupα7 and is probably due to a reduction in the number of ACh binding sites.

This observation can be of fundamental importance, especially when considering the different dupα7 and α7 expression levels in different tissues, and the modulation of their expression by various stimuli (IL-1β, LPS, nicotine, donepezil, HIV-1 gp120) [49,50,53,61], although evidence of the formation of a heteromeric dupα7/α7 receptor in vivo is still missing. Recently, *CHRFAM7A* transgenic mice showed decreased α-Bgtx at the neuromuscular junction and in brain tissue compared to wild-type (WT) animals, showing for the first time that dupα7 expression decreases α7 nAChR ligand binding in vivo [70,71]. Silencing *CHRFAM7A* expression in SH-SY5Y cells significantly increased α7 nAChR-induced neurotransmitter release, which was reduced by the overexpression of the dupα7 protein [58].

The involvement of α7 nAChR in physiological mechanisms, and its association with neurological, neurodegenerative, and inflammatory disorders, confirmed the importance of understanding whether an heteromeric receptor is formed in vivo (i.e., in neurons), and how these isoforms are regulated and eventually deregulated in different pathologies and in response to therapies targeting the cholinergic system.

## 3. CHRNA7 and CHRFAM7A Role in Diseases

### 3.1. Schizophrenia and Neuropsychiatric Disorders

Chromosomal defects (deletion, duplication) and polymorphisms in the 15q13-q14 region are associated with several neuropsychiatric disorders, including schizophrenia [72,73,74], epilepsy [10,36,75,76], psychosis [77,78,79], intellectual disability (ID) [80,81], and autism [35,82]. A direct genetic association of *CHRNA7* with these disorders is controversial; many markers (i.e., the dinucleotide marker D15S1360 in intron 2 of the *CHRNA7* gene and other microsatellite markers) in the vicinity of the *CHRNA7* gene at 15q13-q14, which have a strong association with schizophrenia, have been reported [74]. Other studies have failed to find any genetic linkage to this region [83,84,85,86], thus supporting the idea that schizophrenia is a genetically heterogeneous disease, and that the loci involved may differ depending on ethnicity [87,88]. A stronger linkage with *CHRNA7* was obtained when the P50 auditorily evoked response deficits [89,90], an endophenotype of both schizophrenia and bipolar disorders, and nicotine, transiently normalized the P50 deficit in schizophrenic and bipolar disorder patients, thus confirming the involvement of a nicotinic receptor [91]. The finding that schizophrenic and depressed patients showed the highest prevalence of smoking [92,93] reinforced the hypothesis that smoking may be a form of “self-medication” for these patients [92,94]. Indeed, evidence from animal and human studies have indicated that targeting α7 nAChRs improves cognition, memory, and sensory gating deficits in schizophrenia [95,96,97,98]. It is worth noting that in peripheral blood lymphocytes, *CHRFAM7A* expression is lower in cotinine, in self-reported smokers versus in non-smokers, and is negatively correlated to cotinine (nicotine metabolite) levels, but not with the diagnosis of schizophrenia [99]. Conversely, individuals with two or three copies of the *CHRFAM7A* gene who also smoked showed a higher rate of smoking cessation success under varenicline therapy [67].

The presence of *CHRNA7* and *CHRFAM7A*, in this locus, could increase the probability that the deregulation (gene dosage; altered gene expression) of even one of the two genes could lead to the onset of neuropsychiatric pathologies [55,100,101]. The expression of *α7* is reduced in multiple areas of postmortem brain specimens from schizophrenic patients and from patients with major psychiatric disorders [55,102,103,104,105,106], and 21 polymorphisms have been identified in the promoter region, most of which decrease transcription and are strongly associated with schizophrenia and P50 deficits [34,107,108]. Polymorphisms in both the coding region and the introns that result in splice variants were identified in *CHRNA7* and *CHRFAM7A* genes [109], but none of them were associated with schizophrenia, thus reinforcing the assumption that the receptor is functionally normal and that the causative defect may rely on the expression level of *CHRNA7*. Deletions involving *CHRNA7* are rare but are strongly associated with schizophrenia [72,73]: these patients only have one copy of *CHRNA7* and two copies of *CHRFAM7A*, leading to an altered *CHRNA7/CHRFAM7A* ratio, whereas 15q13.3 duplication has been linked to childhood-onset schizophrenia [110]. A reduced *CHRNA7*:*CHRFAM7A* ratio, due to increased *CHRFAM7A* and/or reduced *CHRNA7* expression, has been reported in the prefrontal cortex of patients with bipolar disorder and schizophrenia [55,59].

Conversely, reduced *CHRFAM7A* expression has been measured in the peripheral blood lymphocytes of schizophrenic patients [111,112] and was determined to be related to illness severity. A recent study [100] examined the expression of *CHRFAM7A* in the peripheral blood lymphocytes of schizophrenic patients, and found a significant negative correlation between *CHRFAM7A* expression and a negative psychopathology score (SANS), but not with a positive score (SAPS). Pairwise analyses before and after antipsychotic treatment revealed an increase in *CHRFAM7A* gene expression during follow-ups compared to the baseline, thus suggesting that *CHRFAM7A* has a role in schizophrenia pathogenesis and treatment. The effect of antipsychotics on *CHRFAM7A* expression has not been found in previous studies [55]. This discrepancy may be explained by limitations in both studies: there is missing knowledge about *CHRNA7* expression (not detected in lymphocytes), the genotype of the analyzed patients in Kalmady’s study, and the use of postmortem human brain samples in Kunii’s analyses, where confounding factors, such as the antemortem use of antipsychotic and antidepressant drugs as well as of substance of abuse, may contribute to the changes observed in the patients in those studies.

Several studies have investigated whether the *Δ2bp* allelic variant of *CHRFAM7A* could confer susceptibility to schizophrenia [8,9,84,113,114,115,116,117,118,119]. Similar to *CHRNA7,* controversial data have been obtained, probably due to differences in ethnic groups, in the phenotype considered (schizophrenia, P50 deficit, antisaccade performance), and in small sample sizes. Sinkus and colleagues [9] reported that despite the deleted *CHRFAM7A* allele being more frequent in Caucasians than in African-Americans, its presence is significantly associated with schizophrenia in both ethnic groups, but the number of *CHRFAM7A* alleles [8], and the presence of at least one copy of the deleted allele [113,117], is sufficient to have a P50 sensory gating deficit. These data have not been confirmed by another study [55], which reported no differences in the allelic frequency of *Δ2bp* and no association with schizophrenia, but instead reported an association with decreased *CHRFAM7A* expression in all subjects, including in the control groups. Moreover, statistical analyses [109] revealed that *Δ2bp* is less frequent in individuals with *CHRNA7* promoter variants [34], which are responsible for decreased *α7* expression and are linked to schizophrenia. Other studies have linked this polymorphism to major depressive disorder [78], bipolar disorder [114], and deficits in episodic memory [116], but not to antisaccade performance [119], the latter two being endophenotypes of schizophrenia.

These studies seem to conclude that, overall, the mutations in the *CHRNA7/CHRFAM7A* “gene cluster” which are associated with neuropsychiatric disorders, fall into the *CHRNA7* promoter, the *Δ2bp* deletion of *CHRFAM7A* and, although very rare, deletion of the *CHRNA7* gene. Moreover, the deregulated expression of both genes (up- and down-regulated) could also contribute to altering the *CHRNA7/CHRFAM7A* ratio, thus contributing to an increased detrimental effect of dupα7 on α7 nAChR function.

### 3.2. Epilepsy and Neurodevelopmental Disorders

The impact of *CHRFAM7A* gene expression and the presence of the *Δ2bp* polymorphism has also been evaluated in several neurological disorders, including in epilepsy and autism spectrum disorders (ASD). A susceptibility locus for the common idiopathic generalized epilepsies (IGEs) has been mapped to the 15q13-q14 region, with a strong association with microdeletions of Chr 15q13.3 [10,36,120,121,122,123,124]. Rozycka et al. [120] determine an association between the *Δ2bp* polymorphism and IGE. In a population-based study, they reported that the frequency of *Δ2bp* carriers was lower in the cohort of IGE patients versus in the cohort of healthy controls, thus providing a protective effect against IGE. This finding is in contrast to what was found in schizophrenic patients [8] and the authors were unable to exclude the possibility that the *Δ2bp* is in linkage disequilibrium (LD) with the true causative polymorphism contributing to IGE [120]. The protective effect could only be explained by considering the negative effect of dupα7 when translated from a wild-type allele, something that is limited in *Δ2bp* allele carriers, and by considering the *CHRFAM7A Δ2bp* allele as a null allele that is unable to be translated into a functional protein. This would lead to an increase in functional α7 receptors.

The study by Rozycka [120] does not report any indication regarding *CHRFAM7A* expression levels. However, independent studies reported a 1.5 Mb microdeletion of the region encompassing the *CHRNA7* gene in a small fraction of IGE patients [121,123,124,125], thus confirming that α7 nAChR may play a fundamental role in IGE pathogenesis. On the other hand, a 15q13.3 duplication involving *CHRNA7* has been implicated as being a risk factor for attention deficit hyperactivity disorder (ADHD) [38]. The presence of CNV and the *Δ2bp* in the *CHRFAM7A* gene, along with the microduplication in the *CHRNA7* gene, has been associated with phenotypic variation in a family with Tourette syndrome, ADHD, and obsessive-compulsive disorders (OCD) [126]. It is worth noting that the expression levels of duplicated genes do not always follow the gene copy number, as shown in brain samples with maternal 15q duplication [127], and for *CHRNA7*, in a neuronal cell model of 15q duplication [128], because of epigenetic alterations that can contribute to defining the *CHRNA7/CHRFAM7A* ratio. Indeed, significantly reduced *CHRNA7* expression has been reported in the frontal cortex of individuals with Rett syndrome or with typical ASD [60]. It was hypothesized that MeCP2 modulates both *CHRNA7* and *CHRFAM7A* expression through long-range chromatin interactions with the 15q11.2–13.3 region, which includes the Prader-Willi/Angelman syndrome’s imprinting center (PWS-IC). MeCP2 deficiency caused by mutations, such as in Rett syndrome [129], or decreased expression, such as in autism samples [130], influences chromatin loop organization, thus leading to altered gene expression.

A lack of *CHRFAM7A* expression in the CD4+ T-lymphocytes of autosomal dominant nocturnal frontal lobe epilepsy (ADNFLE) patients [131], a form of epilepsy that is mainly associated with mutations in other nAChR subunits (*CHRNA2, CHRNA4 and CHRNB2*), suggests that there might be a link between the expression of *CHRFAM7A* and the occurrence of ADNFLE symptoms [131].

### 3.3. Inflammatory Diseases: The “Cholinergic Anti-Inflammatory Pathway”

Neural reflex circuits [132,133] regulate the immune system response to pathogens during the inflammatory process. The central nervous system (CNS) senses inflammation by means of specialized cells in the brain vasculature, choroid plexus, and circumventricular organs as well as through Toll-like receptors (TLRs) and cytokine receptors in the brain. The observation that vagal nerve stimulation attenuated the systemic inflammatory response in rats under endotoxemia, and that the inhibition of the pro-inflammatory cytokines was due to the acetylcholine stimulation of the nicotinic receptors expressed on macrophages, led to the discovery of the “cholinergic anti-inflammatory pathway (CAIP)” [134]. Later studies, reviewed in Hoover, 2017 [33], defined the anatomy of this inflammatory reflex and showed that α7 nAChR plays a major role in the vagal inhibition of the inflammatory response to endotoxemia [32], despite the presence of other nAChRs on immune cells [33,135].

Functional studies on the *dupα7* gene in leukocytes revealed the essential role that it plays in CAIP activation [53]. In the macrophages, where *dupα7* is expressed at higher levels compared to *α7* [49], pro-inflammatory stimuli down-regulate its expression, which is counterbalanced by an increase in *CHRNA7* [49,53], with an NF-κB dependent mechanism. These findings incite the hypothesis that in physiological situations, the high *CHRFAM7A/CHRNA7* ratio allows for the formation of hetero-pentamers due to a large contribution of dupα7, or due to the intracellular retention of α7 [31,50], thus contributing to blocking CAIP activation. In the presence of inflammatory signals, the down-regulation of *CHRFAM7A*, which is paralleled by increased *CHRNA7* expression, will result in the formation of functional α7 nAChRs on the cell membrane and activation of CAIP, subsequently dampening inflammation. Due to pathological status, the altered regulation of both subunits will lead to an altered *CHRFAM7A/CHRNA7* ratio, consequently deregulating CAIP response.

Uncontrolled systemic inflammation leads to sepsis [136], the leading cause of death in intensive care units (ICUs). One of the main causes of its pathogenesis may rely on an imbalance between excessive inflammation and anti-inflammatory mechanisms, and was reviewed in van der Poll et al. 2021 [137]. During an inflammatory challenge imposed by sepsis, *α7* mRNA levels increased during acute illness and returned to the levels found in the controls once the patient recovered [138]. An inverse correlation between the *CHRNA7* expression level and CAIP activity, and disease severity and mortality, was reported in a cohort of patients with sepsis [138] that had no deaths among patients expressing high levels of *CHRNA7*, thus suggesting that measuring the *α7* expression level in PBMC may serve as a prognostic marker for sepsis. A later analysis [64] showed that patients showing a high *CHRFAM7A/CHRNA7* ratio have a poor prognosis compared to patients expressing higher *CHRNA7* levels. A similar conclusion was drawn by Baird and colleagues [139], who showed that *CHRFAM7A* expression was increased, and that *CHRNA7* expression was decreased in Inflammatory Bowel Disease (IBD), thus suggesting that CHRFAM7A may be an unrecognized target for the development of therapeutics for IBD. These data are in accordance with previously reported increases of *CHRFAM7A* caused by the LPS in gut epithelial cells [56], suggesting a different role for dupα7 in this tissue type.

α7 nAChR activation plays a protective role in osteoarthritis (OA), since its absence was associated with severe cartilage degradation in a murine meniscectomy α7 KO model, whereas α7 nAChR activation decreased the IL-1β-induced chondrocyte inflammation [27]. The chondrocytes of OA patients showed elevated *CHRFAM7A* expression [27], which was positively correlated with increased Metalloproteinase-3 (MMP-3) expression, thus suggesting it has a role in the pathophysiology of OA. α7 nAChR silencing, and treatment with specific agonists, revealed a role for α7 nAChR in controlling joint inflammation in a collagen-induced arthritis rat model of rheumatoid arthritis (RA) [140], and in the synovial tissues and fibroblast-like synoviocytes of patients with RA [62]. These tissues express α7and dupα7 receptors, but since there is no information regarding their expression levels in RA compared to in healthy tissues, no conclusions about the role of CHRFAM7A in RA can be drawn.

A protective role for dupα7 has also been proposed. Cerebral ischemia/reperfusion (I/R) injury [141] is an inflammatory-related disorder that is caused by the abnormal activation of the nod-like receptor protein 3 (NLRP3) inflammasome, and microglia pyroptosis, following the disruption to blood supply after an ischemic event [142]. In a case-control study [141], in patients with cerebral I/R injury, the *CHRFAM7A* level was negatively related to the expression of inflammatory cytokines. These data are in accordance with the reported down-regulation of *CHRFAM7A* by inflammatory stimuli [53]. The *CHRFAM7A* overexpression in OGD/R-treated microglia human cell lines, a model of I/R injury, reverted the clinical observation by increasing cell proliferation, decreasing the release of pro-inflammatory cytokines, and promoting M1 to M2 microglia polarization [141], thus suggesting a protective role for *CHRFAM7A* in the attenuation of inflammatory injury of microglia.

*CHRFAM7A* is also down-regulated in hypertrophic scars (HTS) [143]. In a human HTS-like SCID mouse model, transfected CHRFAM7A played a positive role in the amelioration of HTS formation [143] by decreasing TGF-β and CTFG expression and by increasing MMP-1 expression, and conversely to the I/R injury model, attenuating M2 macrophage polarization via the activation of the Notch pathway. An increase in *CHRFAM7A* expression has been measured in radiotherapy-induced lacrimal gland injury, resulting in the inhibition of the p38/JNK signaling pathway and oxidative stress [144].

An imbalanced inflammatory response is responsible for tissue damage and COVID-19 severity [145]. The cholinergic system has been proposed as one of the regulators of COVID-19 induced hypercytokinemia [146,147,148,149,150]. *CHRFAM7A* expression showed a slight tendency to decrease its level in a cohort of SARS-CoV-2 patients compared with healthy controls [14]. When patients were stratified according to disease severity, *CHRFAM7A* expression was significantly lower in critical COVID-19 patients. Again, this is in accordance with inflammatory-induced *CHRFAM7A* down-regulation [53]. Absence of *CHRFAM7A* expression was also associated with increased inflammatory biomarkers, including the C-reactive protein, lymphopenia, extension pulmonary lesions, and plasma viral load, as well as the enhanced expression of genes in the TNF signaling pathway [14]. Ongoing trials are evaluating the usefulness of nicotine to determine the risk of developing the SARS-CoV-2 infection and severe disease; however, these data have led to the hypothesis that individuals with no *CHRFAM7A* duplication, and/or who are carriers of *Δ2bp*, may develop symptomatic and severe COVID-19. This aspect is challenging and deserves further investigation.

All these studies suffer from the lack of *CHRNA7* expression analysis, making it difficult to draw conclusions about the significance of a *CHRFAM7A* increase in I/R injury. It is possible to hypothesize that the increase in *CHRFAM7A* is a step in the inflammation resolution process. In accordance with this hypothesis, is a study on the effect of donepezil [49], an acetylcholine esterase inhibitor (AChEi) used as a symptomatic therapy in AD. Besides its inhibitory effect, it has been proposed as an anti-inflammatory mechanism for this class of compounds [151]. In macrophages, donepezil synergizes with LPS to increase *CHRNA7* expression, resulting in the potentiation of CAIP. The parallel increase in *CHRFAM7A* expression may suggest a role in controlling excessive CAIP activation, with the consequence of an impaired response to inflammatory stimuli [31]. In COVID-19 patients, such deregulation in the cholinergic system may explain severe prognosis. In view of this, CHRFAM7A may be considered as a target for intervention in the fine modulation of CAIP and in the restoration of a homeostatic state.

In individuals with traumatic spinal cord injury (SCI), the presence of the *Δ2bp* polymorphism may affect clinical outcomes [13,152] depending on the severity of the injury. In severe SCI, *Δ2bp* carriers show higher levels of circulating inflammatory molecules (TNF-α, INF-γ, IL-13, CCL11, IL-12p70, IL-8, CXCL10, CCL4, IL-12p40, IL-1β, IL-15 and IL-2), whereas in mild SCI, IL-15 is lower in *Δ2bp* carriers than it is *Δ2bp* non-carriers. Temporal variations in inflammatory mediator analyses have revealed that severe SCI *Δ2bp* carriers had higher levels of IL-8 and CCL2 during the acute phase post-SCI, but no variations were observed in mild SCI. These two molecules are important for macrophage and neutrophil migration and infiltration modulators, as theyplay an important role in the early phases of inflammation post-SCI. Moreover, neuropathic pain was positively associated with IL-13 in *Δ2bp* carriers, but only in severe SCI. Conversely, *Δ2bp* carriers with mild SCI showed that a higher risk of pressure ulcers was positively correlated with the circulating levels of IFN-γ, CXCL10, and CCL4, and negatively associated with IL-12p70 levels.

Animal studies on the association between IL-13 and neuropathic pain produced opposite results [153], thus confirming the importance of including *CHRFAM7A* genotyping in human studies, and that the presence of dupΔα7 can modify the anti-inflammatory function of α7 receptor after SCI and the response to α7 ligands for inflammatory and pain control [154].

#### Other Roles for CHRFAM7A

The higher expression of *CHRFAM7A* in macrophages compared to *CHRNA7* suggests other roles for dupα7 in regulating monocyte/macrophage activation other than the regulation of α7 nAChR activity. The transgenic expression of *CHRFAM7A*, in a model of human systemic inflammatory response syndrome (SIRS) due to severe injury and sepsis, showed an increase in the hematopoietic stem cell (HSC) reservoir in bone marrow, and increased immune cell mobilization, myeloid cell differentiation, and inflammatory monocytes in inflamed lungs [11]. These data confirmed the importance of CHRFAM7A in physiological, pathophysiological, and species-specific responses, and point to its role as a doubled-edge sword, being both protective and detrimental in an emergency myelopoiesis model upon injury. Indeed, limiting HSC exhaustion upon injury can increase the responsiveness to a second stimulus, thus improving, for example, survival.

The overexpression of *CHRFAM7A* in monocyte-like cell lines has shown reduced cell migration, chemotaxis to the monocyte chemo-attractant protein (MCP-1), and the inhibition of anchorage-independent colony formation [155]. The mechanism of this effect is still unknown and is in contrast with previous data from the same group [11,54], which reported that the increased expression of gene encoding for proteins favors cell migration, adhesion, and the formation of cell clusters upon *CHRFAM7A* overexpression. Other affected pathways tied to pro-inflammatory conditions [156,157] include Type L interferon and cancer pathways [54]. Unexpectedly, *CHRFAM7A* overexpression increased the CHRNA7 level [54], resulting in increased α-Bgtx binding. This is in contrast with previously discussed data [50,52,63,64], but it corroborates the finding that *CHRNA7* and *CHRFAM7A* expression are co-regulated [49,53,56]. Thus, we cannot rule out that many of the direct effects of *CHRFAM7A* overexpression in the modulation of different signaling pathways are indeed the result of the modulation of the metabotropic function of α7 nAChR [158].

### 3.4. Neurodegenerative Diseases

Epidemiological [159] and in vitro studies indicate a protective role of nicotine assumption in Alzheimer’s (AD) and Parkinson’s (PD) disease against glutamate and β-amyloid-related cytotoxicity [160], and on dopaminergic neurons via an anti-inflammatory mechanism that is mediated by the modulation of microglial activation and astrocyte apoptosis [161,162]. α7 nAChR is expressed by microglial cells, where it plays important roles in controlling their activation [163,164], and several reports support the existence of a brain CAIP [18,165,166].

One of the hallmarks of AD and PD is the loss of cholinergic neurons and a selective decrease in the number of nicotinic receptors (reviewed in [160,167]) that is associated with an inflammatory state, which is generally caused by the hyper-activation of the microglia [168,169]. For these reasons, the cholinergic system is one of the therapeutic targets in AD. Acetylcholinesterase inhibitors (AChEi), such as galantamine or donepezil, are indeed effective, and ameliorate the cognitive symptoms of AD [170].

Genetic variations in the *CHRNA7* gene, or the altered expression and function of α7 nAChR have been associated with AD [41,160,171,172] and PD [173,174], and in the case of AD, the reduced α7 expression is correlated with beta-amyloid (Aβ) plaque deposition and cognitive impairments [41,160,175]. The high-affinity binding of Aβ oligomers to α7 nAChR shows Aβ concentration-dependent opposite effects. At physiological concentrations (pM to low nM range), Aβ triggers a change in α7 to a desensitized conformation that still responds to agonists, while at higher concentrations (nM to low μM range), Aβ acts as a negative modulator [176,177,178,179]. This results in the activation of neuroprotective (pM range) or neurotoxic (nM range) signaling pathways [167,177,180,181]. Moreover, Aβ-α7 interaction leads to intraneuronal accumulation of Aβ_1–42_ and Aβ-induced tau protein phosphorylation [182,183].

*CHRFAM7A* CNV is over-represented in mild cognitive impairment (MCI) and late-onset AD patients [184,185,186]. Ordered-subset analysis (OSA) by age at AD onset confirmed *CHRFAM7A* as one of the loci that can confer the risk of AD [187] in a subset of patients. Here, reduced *CHRFAM7A* expression in human post-mortem temporal lobe tissue was correlated with the disease state (AD vs. controls) and with CNV [187]. In view of this, the increased functional α7 nAChR due to lower *CHRFAM7A* expression is able to sustain enhanced Aβ_1–42_ internalization and neuronal vulnerability, and therapies with α7 antagonists may be more effective in these AD patients.

The presence of the *Δ2bp* polymorphism did not confer susceptibility to Alzheimer’s disease in a case-control study [188]. Patients were stratified by three different *CHRFAM7A* genotypes (genotype 1: two wild-type alleles; genotype 2: one wild-type allele and one *Δ2bp* allele; genotype 3: two *Δ2bp* alleles) in four types of dementia, including AD, dementia with Lewy’s bodies (DLB), Pick’s disease (PiD), and vascular dementia (VD). The results showed that the *CHRFAM7A* genotype with two wild-type copies of the gene was over-represented in the AD, DLB, and PiD cases, but not in VD [189], which may be due to the heterogeneity of the underlying pathology [190]. This observation indicates that the *Δ2bp* polymorphism, as a protective factor against the development of these forms of dementia, which are associated with abnormal protein aggregations, and again, we have to hypothesize that this may occur if *dup*Δ*α7* behaves as a null allele, results in increased α7 nAChR assembly. The protective role of nicotine in AD and PD reinforces this hypothesis since nicotine, apart from stimulating α7 nAChR, down-regulates the mRNA of *CHRFAM7A* [50].

These apparently controversial results may depend on multiple confounding factors, such as the age of onset, the duration of the disease from diagnosis, and the type of disease.

However, therapeutic approaches aimed at both stimulating or antagonizing α7 nAChR failed to translate into humans [191].

By using median ganglionic eminence (MGE) neuronal progenitors, obtained by the neuronal differentiation of human induced pluripotent stem cells (iPSC) from AD patients with 0 or 1 copies of *CHRFAM7A*, it has been shown [12,192] that dupα7 decreases the probability of the α7 nAChR channel opening in the presence of the positive allosteric modulator (PAM) PNU 120,596 (PNU) with currents that desensitize faster [192], and this desensitization increases as a function of the *CHRFAM7A* dosage. These data suggest that the number of *CHRFAM7A* copies may affect the response to α7 PAM, agonists, and antagonists in different individuals. Moreover, dupα7 mitigates Aβ_1–42_ uptake at a higher concentration, as well as the α7-dependent Aβ-induced inflammatory response, suggesting a protective role in AD during the Aβ_1–42_ accumulation phase. At higher Aβ concentrations, the presence of dupα7 increases IL-1β and TNF-α release as an ultimate signal for the activation of an anti-inflammatory response. In this sense, the absence of *CHRFAM7A* is a risk factor for AD [192]. In this iPSC model, *dup*Δ*α7* behaves as a null allele [12], as its expression does not affect PNU-modulated α7 nAChR currents and Aβ_1–42_ uptake.

Of greater interest, is that dupα7 expression affects the response to AChEi (donepezil and rivastigmine) and encenicline, a selective α7 agonist used in AD therapy [12]. In *CHRFAM7A* non-carriers, the drugs reduced Aβ_1–42_ uptake, and Aβ_1–42_ induced toxicity and apoptosis. On the other hand, in *CHRFAM7A* carrier cells, the neurotoxicity caused by donepezil and encenicline was unchanged, and donepezil increased apoptosis. These data suggest that *CHRFAM7A* may affect drug response.

Non-carriers of the functional *CHRFAM7A* allele comprise 25% of the human population, whereas 75% are carriers, with no differences being observed between normal aged controls and AD patients, thus suggesting that *CHRFAM7A* is not associated with a disease phenotype, which is in contrast with the data discussed above [184,185,186,187,189]. A double-blind pharmacogenetics study [12] showed that non-carriers of the functional *CHRFAM7A* allele had a superior response to the AChEi, which is in agreement with the iPSC model, and this lasted over a 7-year observation period. The *CHRFAM7A* carriers did not demonstrate a treatment response effect.

To demonstrate the role of *CHRFAM7A* in Aβ-associated neuroinflammation, the same group [193] showed that dupα7 also mitigates Aβ_1–42_ uptake in iPSC-derived microglial-like cells, but unlike the iPSC-derived neurons, the presence of CHRFAM7A induced a high NF-κB-mediated innate immune response by prolonging its nuclear presence, resulting in microglia activation. In these cells, nicotine paradoxically increases the proinflammatory response to LPS in *CHRFAM7A* carrier cell lines, thus suggesting that the presence of dupα7 antagonizes the anti-inflammatory role of homomeric α7. These data are in line with earlier results obtained from the overexpression of *CHRFAM7A* in RAW264.7 murine macrophages [64], but not with the findings of other groups [141,143]. A limit of these studies is that most of the conclusions rely on data obtained from comparing a *CHRFAM7A* non-carrier cell line and its isogenic counterpart, which expresses *CHRFAM7A* under the control of a constitutive promoter. LPS, via an NF-κB dependent mechanism, and nicotine, down-regulate *CHRFAM7A* expression [50,53], an aspect that might lead to different conclusions, and it would be interesting to know whether Aβ_1–42_ plays a similar role. Similar data on the LPS/nicotine synergism obtained in a native *CHRFAM7A* carrier cell line [193] may suffer from long-term LPS treatment (24 h), as it has been shown that 3–6 h of LPS down-regulates *CHRFAM7A* expression [53] that returns to basal level after 24 h, but no data on *CHRFAM7A* expression have been presented. On the other hand, long-term exposure to LPS may resemble a chronic inflammatory insult, such as that caused by Aβ in AD, where an excessive inflammatory insult may lead to CAIP dysregulation.

Proteomic profiling in a *CHRFAM7A* transgenic mouse brain [71] showed that dupα7 overexpression modulates the expression of proteins involved in the α7 nAChR signaling pathways, and that they are related to the pathogenesis of neurological and neuropsychiatric disorders, including Parkinson’s and Alzheimer’s disease. In particular, most of these genes are mitochondrial components, and are involved in the maintenance of the anti-oxidative response, whose deregulation is one of the main causes of neurodegenerative and neuropsychiatric disorders.

The HIV-1 (human immunodeficiency virus type-1) envelope protein gp120_IIIB_ induces α7 nAChR in neuronal cells and in the brain, particularly in the striatum, the basal ganglia’s primary input, and promotes cell death in a calcium-dependent manner, an effect abrogated by α7 antagonists [194]. Conversely, *CHRFAM7A* is down-regulated in some patients [61], thus increasing the *CHRNA7/CHRFAM7A* ratio. α7/dupα7 deregulation may thus play a role in the development of HIV-1-associated neurocognitive disorders (HAND) [61].

### 3.5. Cancer

The dysregulation of nAChRs gene expression and/or the dysfunction of these receptors may be involved in the development of lung tumors [45]. α7 nAChR is overexpressed in squamous cells, but not in adenocarcinoma lung tumors, and is a determinant for the progression of this type of cancer. Nicotine further increases *CHRNA7* expression and function, mediating its proliferative, survival, and angiogenic effects. In both cell types, increased *α5* and *β4* accompanies *dupα7* and *β3* down-regulation. In particular, reduced *CHRFAM7A* expression may facilitate the α7-mediated oncogenic process by relieving the negative effect of dupα7. Indeed, its overexpression in two NSCLC cell lines (A549 and SK-MES-1) blocked nicotine- or NKK-induced tumor progression, both in vitro and in vivo, in an athymic mouse model implanted with A549dupα7 or A549 xenografts [195]. These data suggest that dupα7 can be considered as a therapeutic target in smoking-related cancers.

## 4. Conclusions

Ever since its discovery in 1998, understanding the nature and function of the *CHRFAM7A* gene has become a great challenge to respond to the failure of translating α7-directed therapeutic approaches into humans, with treatments aimed at both stimulating or antagonizing α7 nAChR. Its assembly with the α7 subunit leads to the formation of heteromeric receptors, which show reduced ion conductance and a reduced affinity for α-Bgtx, thus suggesting a negative role of dupα7 on α7 nAChR function.

The great amount of data on *CHRFAM7A* involvement in neurological and non-neurological disorders reported here, and summarized in Table 2, showed a contrasting effect, being protective in some cases and related to a poor prognosis or a susceptibility gene in the pathogenesis of that disorder in others, but these studies have highlighted the impact of this gene in a variety of α7 nAChR-related disorders. These confounding results may suffer from being clinical studies based on small sample sizes and from the lack of expression levels detected from both *CHRNA7* and *CHRFAM7A* and/or *CHRFAM7A* genotyping. Most importantly, the *CHRFAM7A* copy number and the presence of the *Δ2bp* polymorphism influence the quality of protein products, and therefore, the possible effect on α7 nAChR. Moreover, dupα7 can play a different role depending on the tissue type, thus shifting the ionotropic feature of α7 nAChR to that of a metabotropic receptor, as indicated by the modulation of signaling pathways upon *CHRFAM7A* overexpression. One important issue to take into account is that the expression levels of *dupα7* and *α7* vary in different tissues and can be modulated by various stimuli. The deregulated expression of both genes (up- and down-regulated) thus contribute to altering the *CHRNA7/CHRFAM7A* ratio, paving the way to an increased detrimental effect of dupα7 on α7 nAChR function.

Many questions are still pending; in particular, the use of the *CHRFAM7A* expression level, which can be measured via blood samples, as a prognostic marker even in neurological disorders, or as therapeutic target, but whether to reduce or increase its expression, remains to be elucidated. On the other hand, computational model studies will be very important for the design of drugs targeting heteromeric receptors.

## Figures and Tables

**Figure 1 ijms-23-03463-f001:**
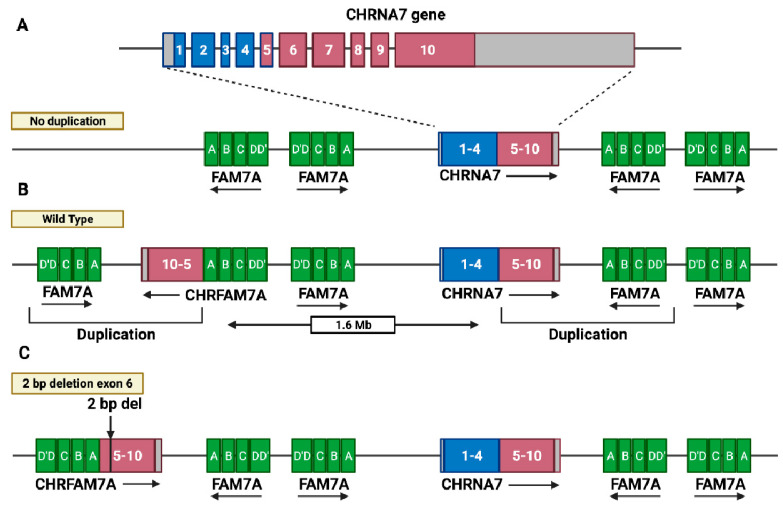
The *CHRNA7* and *CHRFAM7A* locus on Chr. 15q13.3. (**A**) Scheme of the genomic structure of the ten-exon *CHRNA7* gene, flanked by four FAM7A cassettes. Numbers indicate exons (1 to 4 in blue, 5 to 10 in red). (**B**) *CHRNA7* exons 5 to 10 (red) are duplicated and fused in-frame with one *FAM7A* cassette (green), 1.6 Mb from, and in the opposite orientation to, the *CHRNA7* gene, giving rise to the hybrid *CHRFAM7A* gene (wild-type allele). (**C**) A deletion of two base pairs in exon 6 (2 bp del), generating the *Δ2bp* polymorphic allele, which is associated with *CHRFAM7A* gene inversion with the same orientation as the *CHRNA7* gene. (Created with BioRender.com (accessed on 23 January 2022)).

**Figure 2 ijms-23-03463-f002:**
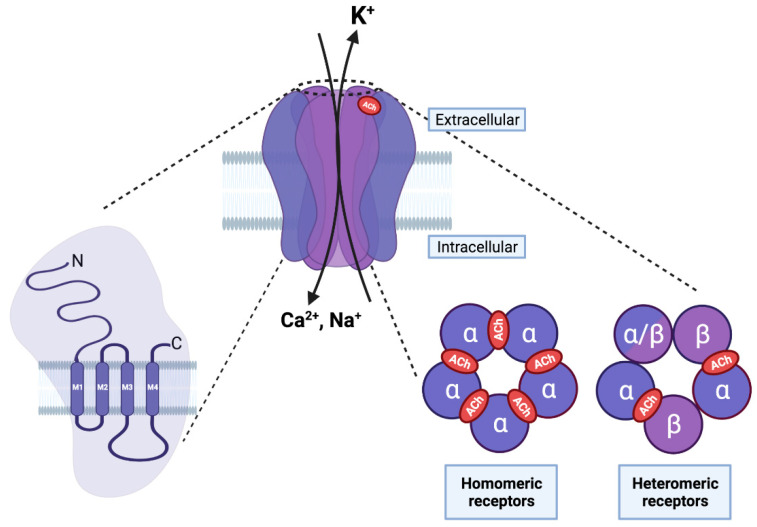
Structure of nAChRs. On the left, each nAChR subunit is composed of an extracellular amino terminal portion, followed by three hydrophobic transmembrane domains (M1–M3), a large intracellular loop, a fourth transmembrane domain (M4), and an extracellular carboxy–terminus. In the middle, the pentameric arrangement of nAChR subunits is shown in an assembled receptor. The M2 transmembrane domain of the five subunits forms the central pore and possesses amino acids that are important for ion selectivity, permeability, and channel gating. On the right, five subunits can assemble to form homo- (five α subunits) or hetero-pentameric (α and β subunits) receptors. The orthosteric ligand binding site is formed between two α subunits (in red) in homomeric receptors, and between the α and β subunits in an heteromeric receptor. (Created with BioRender.com (accessed on 23 January 2022)).

**Figure 3 ijms-23-03463-f003:**
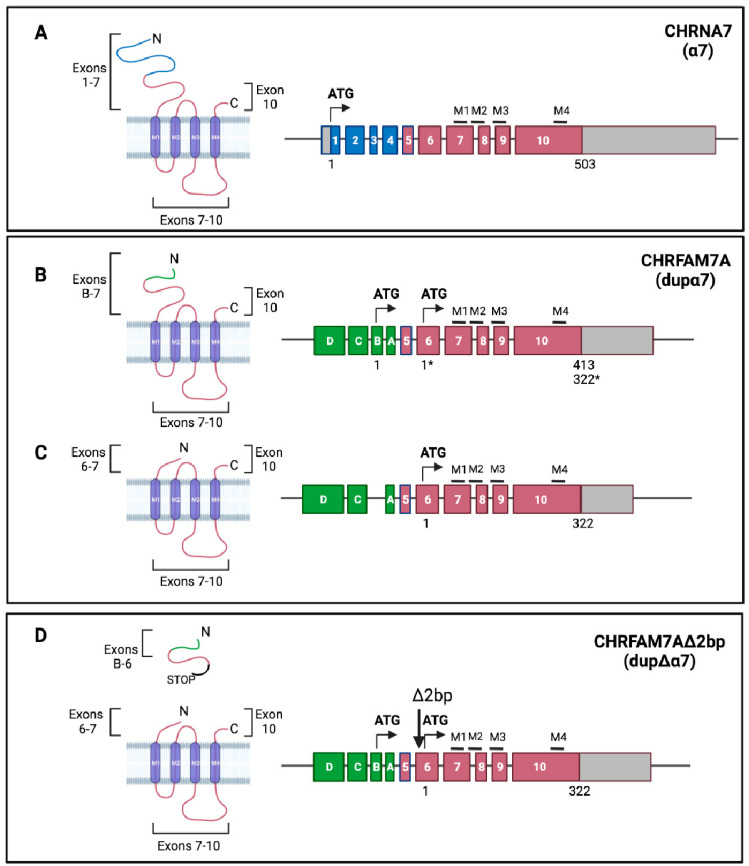
The α7 and dupα7 receptor. Relative protein products of *CHRNA7* and *CHRFAM7A*. (**A**), *CHRNA7* encodes the 503 aa α7 subunit: exons 1–7 encode the extracellular N-terminus domain, including the signal peptide and the acetylcholine binding domain; exons 7–10 encode the M1–M3 transmembrane domains, the long M3–M4 intracellular loop, the M4 transmembrane domain, and the small extra-cellular C-terminal domain. (**B**,**C**), *CHRFAM7A* encodes the dupα7 subunit via two alternatively spliced mRNAs that differ in terms of the presence of exon B. Translation starting from exon B (**B**) results in a 413 aa protein that differs from α7 for the N-terminal domain, and a 27 amino acid-long peptide that is encoded by ExB–Ex6. Translation starting from exon 6 (**C**) results in a 322 aa protein that can be considered a truncated α7 subunit with a short N-terminus encoded by exons 6–7. (**D**), *CHRFAM7AΔ2bp.* The presence of the *Δ2bp* polymorphism (-/TG) in exon 6 results in a short, truncated protein that is encoded by exons B–6 because of the insertion of a translation stop codon and a receptor encoded by exons 6–10 (dupΔα7), due to the translational start site in exon 6. This protein is similar to the one obtained in (**C**). (Created with BioRender.com (accessed on 23 January 2022)).

**Figure 4 ijms-23-03463-f004:**
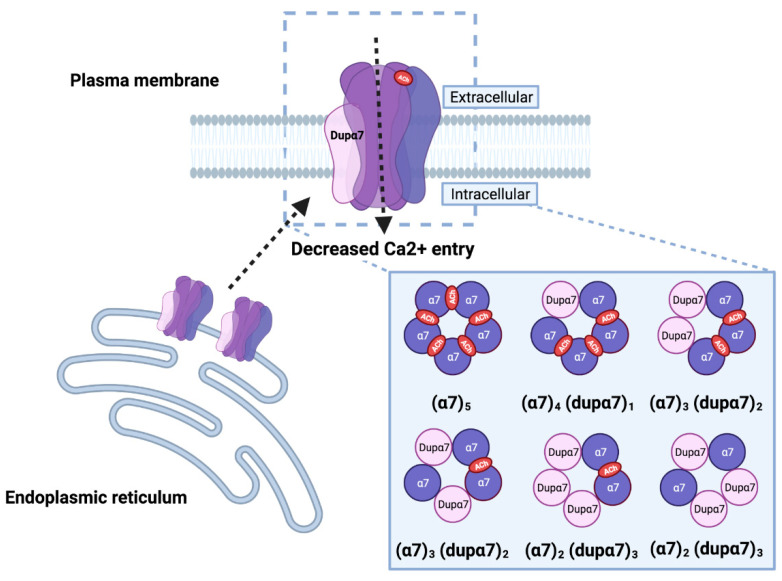
The heteromeric α7/dupα7 receptor. Dupα7 can assemble and interact with α7 subunits to form heteropentameric receptors, causing reduced function of the receptor at the cell membrane (dashed line), as demonstrated by a net reduction in the macroscopic current, that is inversely proportional to the amount of dupα7 and/or retention (dashed line) at the endoplasmic reticulum (ER). In order for the heteromeric receptor to be functional, it should include at least two α7 subunits to form the agonist binding site. The possible stoichiometries of the heteromeric receptor are the result of experimental data and computational model simulations. (Created with BioRender.com (accessed on 23 January 2022)).

**Table 1 ijms-23-03463-t001:** Allelic frequency of 2bp deletion (*-/TG*) from the 1000 Genomes Project. EAS = East Asian; EUR = European; AFR = African; AMR = Ad-mixed Americans; SAS = South Asian.

Sample		Alleles	
Population	Chromosome Sample Count	-	CA
EAS	1008	0.66469997	0.33530000
EUR	1006	0.37770000	0.62230003
AFR	1322	0.08170000	0.91829997
AMR	694	0.44090000	0.55910003
SAS	978	0.48469999	0.51530004

**Table 2 ijms-23-03463-t002:** *CHRFAM7A* involvement in neurological and non-neurological disorders. ↓ Reduced expression; ↑ increased expression; =, unchanged; n.d., not determined; und, undetectable. AD, Alzheimer’s disease; DLB, Dementia with Lewy’s bodies; PiD, Pick’s disease; PD, Parkinson’s disease; HAND, HIV-1-associated neurocognitive disorder; HSC, hematopoietic stem cell; NSCLC, non-small cell lung carcinoma. Numbers in square brackets indicate references.

Disease	Expression Level/Genotype		Biological Effect	Therapeutic Intervention
	*CHRNA7*	*CHRFAM7A*		
** *Schizophrenia and neuropsychiatric* ** ** *disorders* **				
Postmortem brain sample from schizophrenic and psychiatric patients [55,59,72,73,102,103,104,105,106,109]	↓	↑	Decreased *CHRNA7/CHRFAM7A* ratio	
Childhood-onset schizophrenia [110]	Gene duplication	n.d.	Increased *CHRNA7/CHRFAM7A* ratio	
PBMC of schizophrenic patients [100,111,112]	n.d.	↓	Positive correlation with illness severity; negative correlation between *CHRFAM7A* expression and negative psychopathology score (SANS), but not with a positive score (SAPS)	Antipsychotics increase *CHRFAM7A* expression [100]
PBMC of schizophrenic patients [99]	und	↓	Lower CHRFAM7A expression in smokers, not associated with diagnosis	
Association studies [9,113,117]	2 copies [9,113] n.d. [117]	*Δ2bp*	P50 sensory gating deficit	
Association studies [55]	=	*Δ2bp*	No differences in *Δ2bp* allele frequency between ethnic groups; association with reduced *CHRFAM7A* expression in patients as well as in control group	
Major depressive and bipolar disorders, deficit in episodic memory [78,114,116]	n.d.	*Δ2bp*	SNP associated with listed disorders	
Antisaccade performance [119]	n.d.	*Δ2bp*	No association	
** *Epilepsy and Neurodevelopmental* ** ** *disorders* **				
Idiopathic generalized epilepsies (IGEs) [120]	2 copies	*Δ2bp*	Frequency of *Δ2bp* carriers was lower in IGE patients’ cohort versus healthy controls	
Idiopathic generalized epilepsies (IGEs) [121,123,124,125]	1.5 Mb microdeletion	n.d.	Role in IGE pathogenesis	
Genetic generalized epilepsy [36]	Missense mutations	*Δ2bp*	No association with *Δ2bp* allele; association with missense mutations in *CHRNA7*	
Attention Deficit Hyperactivity Disorder (ADHD) [38]	15q13.3 duplication	n.d.	Duplication involves *CHRNA7*, considered a risk factor	
Tourette Syndrome, ADHD, and obsessive compulsive disorders (OCD) [126]	Microduplication	CNV and *Δ2bp*	Phenotypic variation in a family with all the listed disorders; altered *CHRNA7/CHRFAM7A* ratio	
Rett syndrome [60]	↓	n.d.	It was hypothesized that MeCP2 modulates both *CHRNA7* and *CHRFAM7A* expression by epigenetic modifications	
Autosomal dominant nocturnal frontal lobe epilepsy (ADNFLE) [131]	n.d.	Not expressed	CHRFAM7A is expressed in PBMC of healthy individuals but not in ADNFLE patients, suggesting it can be an important factor in ADNFLE pathogenesis	
** *Inflammatory* ** ** *diseases* **				
Sepsis [138]	High	High/low	Altered *CHRNA7/CHRFAM7A* ratio: high *CHRFAM7A/CHRNA7* ratio has a poor prognosis compared with patients expressing higher *CHRNA7* levels	prognostic marker
Inflammatory Bowel Disease (IBD) [139]	↓	↑		dupα7 down-regulation
Osteoarthritis (OA) [27]	=	High/low	CHRFAM7A expression correlates with MMP-3 and MMP-13 mRNA level in human OA chondrocytes	α7 agonists
Rheumatoid arthritis (RA) [62]	Expressed	Expressed	α7 silencing/α7 agonists reveal a role for CHRNA7 in controlling joint inflammation. Both isoforms are expressed in synovial tissues from RA patients, but no conclusions on the role of CHRFAM7A can be drawn due to the missing healthy controls	
Cerebral ischemia/reperfusion (I/R) injury [141]	n.d.	↓	dupα7 has a protective role; its expression is negatively related to the expression of inflammatory cytokines	dupα7 up-regulation (see OGD/R microglia cells model)
Hypertrophic scars (HTS) [143]	n.d.	↓	Its expression ameliorates HTS formation	See animal model
Radiotherapy-induced lacrimal gland injury [144]	n.d.	↑	Inhibition of the p38/JNK signaling pathway and oxidative stress	
COVID-19 [14]	und	↓	Reduction correlates with disease severity	
Spinal cord injury (SCI) [13,152]	n.d.	*Δ2bp*	Its presence may affect clinical outcomes. Positively correlated with high levels of inflammatory molecules in severe SCI	
** *Neurodegenerative diseases* **				
MCI and late-onset AD patients [184,185,186,187]	n.d.	↓	dupα7 has a protective role as the increased functional α7nAChR, due to lower *CHRFAM7A* expression, will sustain enhanced Aβ_1-42_ internalization and neuronal vulnerability	α7 antagonists dupα7 up-regulation
AD, DLB and PiD [189]	n.d.	wt allele over-represented	*Δ2bp* polymorphism as a protective factor	
AD: association study [12]	2 copies	75% carriers 25% non-carriers	No differences between normal aged and AD patients in the *CHRFAM7A* carriers, thus suggesting that *CHRFAM7A* is not associated with the disease phenotype. *CHRFAM7A* influences drug response to AChEi, as non-carriers showed higher response to AChEi over a 7-year follow-up	
HIV-1 [61,194]	↑	↓	Increased *CHRNA7/CHRFAM7A* ratio. dupα7 may play a protective role against the development of HAND	dupα7 up-regulation
** *Cancer* **				
Squamous cell lung tumor [45]	↑	↓	dupα7 has a protective role, as its down regulation facilitates the oncogenic properties of α7	dupα7 up-regulation
** *Cell and animal models* **				
LPS stimulated macrophages [49,53]	↑	↓	Activation of CAIP	
*CHRFAM7A* transgenic mice [70,71]	=	Over-expression	Decrease in α7 nAChR ligand binding at the neuromuscular junction	
*CHRFAM7A* silencing in SH-SY5Y cells [58]	=	Over-expression	Increased α7 nAChR induced neurotransmitter release	
LPS stimulated gut epithelial cells [56]	Small changes	↓	CHRFAM7A/CHRNA7 ratio increase varies depending on gut epithelial cell line	
OGD/R-treated microglia human cell lines (model of I/R injury) [141]	n.d.	Over-expression	Attenuation of microglia mediated-inflammatory response by increased cell proliferation, decreased pro-inflammatory cytokines, and promotion of M1 to M2 microglia polarization	
Human HTS-like SCID mouse model [143]	n.d.	Over-expression	Positive role in the amelioration of HTS formation by decreasing TGF-β and CTFG expression and increasing MMP-1 expression	
Donepezil-treated human macrophages [49]	↑	↓	The increased *CHRFAM7A* expression suggested a role in controlling excessive CAIP activation	
*CHRFAM7A* transgenic model of human systemic inflammatory response syndrome (SIRS) [11]	n.d.	Over-expression	Increased HSC reservoir, increased immune cell mobilization, myeloid cell differentiation	
Monocyte-like cells (THP-1) [54,155]	↑	Over-expression	Reduced cell migration and chemotaxis to monocyte chemo-attractant protein (MCP-1); inhibition of anchorage-independent colony formation; increased α-Bgtx binding	
Medial ganglionic eminence (MGE) neurons derived from iPSCs from AD patients [12,192]	2 copies	1 copy	dupα7 mitigates Aβ_1–42_ uptake at a higher concentration, and a α7-dependent Aβ-induced inflammatory response, suggesting a protective role in AD during the phase of Aβ_1–42_ accumulation. The absence of *CHRFAM7A* suggests a risk factor in AD. On the other hand, dupα7 expression affects the response to AChEi	
AD patients’ iPSC-derived microglial-like cells [193]	2 copies	1 copy	dupα7 mitigates Aβ_1–42_ uptake and induces a high NF-κB-mediated innate immune response, resulting in microglia activation. Nicotine increases the proinflammatory response to LPS in *CHRFAM7A* carrier cell lines, thus suggesting that the presence of dupα7 antagonizes the homomeric α7 anti-inflammatory role	
*CHRFAM7A* transgenic mice brains [71]	**=**	Over-expression	Modulation of the expression of proteins involved in α7 nAChR signaling pathways, and related to the pathogenesis of neurological and neuropsychiatric disorders, such as PD and AD, including anti-oxidative pathways	
NSCLC cell lines (A549, SK-MES-1) [195]	=	Over-expression	Blocking nicotine- or NKK-induced tumor progression, in an athymic mouse model implanted with A549dupα7 or A549 xenografts	

## Data Availability

Not applicable.

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
