# Peer review of "The Human-Restricted Isoform of the α7 nAChR, CHRFAM7A: A Double-Edged Sword in Neurological and Inflammatory Disorders"

_ijms, 2022, doi:10.3390/ijms23073463_

Round 1
Reviewer 1 Report
The figures are very nicely done and the authors have done an excellent job in collating comprehensive information on CHRFAM7A.
However, the paper requires major restructing and condensing of sections. It is too lengthy and there are lots of grammatical problems. Please organize the sections clearly. In your introduction clearly state why is this review important and what is the take home message readers will get from this review.
It is not possible to identify all the grammar and other cosmetic issues but here are few examples:
CHRFAM7A is a relatively recent and exclusively human gene arisen from the partial du- 12 plication of exon 5 to 10 of the α7 neuronal nicotinic acetylcholine receptor subunit (α7 nAChR) encoding gene, CHRNA7. Should be arising grammatically but in general, there are other ways to represent this sentence. Please rephrase.
In the presence of CHRNA7 gene product, dupα7 can assemble and form heteromeric receptors that in order to be functional, should include at least two α7 subunits to form the ag onist-binding site. Please remove extra space in the word agonist.
CHRFAM7A (dupα7) is a human-lineage specific gene [1-3], appeared about 3.5 million 34 years ago during evolution, as confirmed by the absence in other primates and the absence in rodents excludes the hypothesis of gene loss [5]. This sentence does not read well. Please rephrase.
CHRNA7 AND CHRFAM7A: WHAT ROLE IN DISEASES? This heading reads odd.
Conversely, a reduced CHRFAM7A expression has been measured in peripheral blood lymphocytes of schizophrenic patients [112,113], related to illness severity. A recent study [100]- This reference appears as bold. Please correct it.
Author Response
The Authors are very grateful to the Reviewer for its thoughtful and stimulating review of the manuscript and we hope that the content of the revised version, following Reviewer’s indications, results substantially improved.
Q: The figures are very nicely done and the authors have done an excellent job in collating comprehensive information on CHRFAM7A.
R: Thanks
Q: However, the paper requires major restructing and condensing of sections. It is too lengthy and there are lots of grammatical problems. Please organize the sections clearly.
R: We are aware that the manuscript could be long, but this is the first review that summarizes all the studies on CHRFAM7A. We might have shortened the part on the biochemical studies, but we wanted to let the readers understand the big effort done in order to understand whether a heteromeric receptor is indeed formed, and the many still opened questions to address. Regarding the part on the role in different diseases, we added a summary in form of a table (Table 2), to give the readers an at-a-glance summary of the many different role of CHRFAM7A in different pathologies. We hope that this may encounter the reviewer’s request.
Q: In your introduction, clearly state why is this review important and what is the take home message readers will get from this review.
R: The introduction has been implemented with sentences aimed at highlighting the importance of the review and the take home message. The new part is red-typed. We hope to have addressed the reviewer’s request.
Q: It is not possible to identify all the grammar and other cosmetic issues but here are few examples:
R: The IJMS language editing service has extensively revised the manuscript. All the corrections are red-typed
CHRFAM7A is a relatively recent and exclusively human gene arisen from the partial du- 12 plication of exon 5 to 10 of the α7 neuronal nicotinic acetylcholine receptor subunit (α7 nAChR) encoding gene, CHRNA7. Should be arising grammatically but in general, there are other ways to represent this sentence. Please rephrase.
R: It has changed in "arising", as suggested by the IJMS language editing service
Q: In the presence of CHRNA7 gene product, dupα7 can assemble and form heteromeric receptors that in order to be functional, should include at least two α7 subunits to form the ag onist-binding site. Please remove extra space in the word agonist.
R: Fixed
Q: CHRFAM7A (dupα7) is a human-lineage specific gene [1-3], appeared about 3.5 million 34 years ago during evolution, as confirmed by the absence in other primates and the absence in rodents excludes the hypothesis of gene loss [5]. This sentence does not read well. Please rephrase.
R: The sentence has been rephrase according to the suggestions by the IJMS english-editing service. We hope that in this form it results clearer to the reader.
Q: CHRNA7 AND CHRFAM7A: WHAT ROLE IN DISEASES? This heading reads odd.
R: We changed the heading into “CHRNA7 and CHRFAM7A role in diseases”
Q: Conversely, a reduced CHRFAM7A expression has been measured in peripheral blood lymphocytes of schizophrenic patients [112,113], related to illness severity. A recent study [100]- This reference appears as bold. Please correct it.
R: Fixed
Reviewer 2 Report
Di Lascio and co-authors review current research conducted on CHRFAM7A, a human gene arisen from the partial duplication of the α7 nicotinic acetylcholine receptor subunit (α7 nAChR) encoding gene. The authors summarize current knowledge about the biophysics of α7 and dupα7, i.e., the protein product of the CHRFAM7A gene. They compile information about the involvement of α7 and dupα7 in different pathologies, e.g., schizophrenia, epilepsy, inflammation, and neurodegenerative disorders, to provide evidence that α7 and dupα7 play critical roles in these diseases.
The review is very informative, well structured, and nicely written. I very much enjoyed reading it. The authors could consider to condense all information about the pathology of α7 and dupα7 in form of a table. Readers will appreciate to have an at-a-glance summary of this.
Minor Comments:
Line 59: Box 1 shows only five possible combinations of the CHRFAM7A genotype, but in the text, it is stated that there are six genotypes.
Lines 70 and 150: Code for Δ2bp polymorphism is not consistently used. Is it (-/TG) or (-/CA)?
Line 163: “may putatively originates” should be “may putatively originate”
Line 271: “in the contest” should be “in the context”
Line 306: “this isoforms are regulated” should be “these isoforms are regulated” (Plural)
Line 397: “to what found” should be “to what was found”
Line 407: “has been implicate” should be “has been implicated”
Line 443: “pro-inflammatory stimuli down-regulates” should be “pro-inflammatory stimuli down-regulate” (Plural)
Line 444: “by an NF-kB dependent mechanisms” should be “by an NF-kB dependent mechanism” (Singular)
Line 581: “one of the therapeutic target” should be “one of the therapeutic targets” (Plural)
Author Response
Q: Di Lascio and co-authors review current research conducted on CHRFAM7A, a human gene arisen from the partial duplication of the α7 nicotinic acetylcholine receptor subunit (α7 nAChR) encoding gene. The authors summarize current knowledge about the biophysics of α7 and dupα7, i.e., the protein product of the CHRFAM7A gene. They compile information about the involvement of α7 and dupα7 in different pathologies, e.g., schizophrenia, epilepsy, inflammation, and neurodegenerative disorders, to provide evidence that α7 and dupα7 play critical roles in these diseases.
The review is very informative, well structured, and nicely written. I very much enjoyed reading it. The authors could consider to condense all information about the pathology of α7 and dupα7 in form of a table. Readers will appreciate to have an at-a-glance summary of this.
R: The Authors are very grateful to the Reviewer for its thoughtful and stimulating review of the manuscript and we hope that the content of the revised version, following Reviewer’s indications, results substantially improved.
In particular, we summarized all the information of the role of CHRFAM7A in diseases in Table 2, as suggested, and we hope that this results in an informative at-a-glance summary.
All the minor comments have been addressed has reported, and signed in the text (red-typed).
Moreover, the manuscript has been extensively revised for english by the IJMS language editing service.
Minor Comments:
Q: Line 59: Box 1 shows only five possible combinations of the CHRFAM7A genotype, but in the text, it is stated that there are six genotypes.
R: The missing sixth genotype is now shown. It was a “cut and paste” mistake.
Lines 70 and 150: Code for Δ2bp polymorphism is not consistently used. Is it (-/TG) or (-/CA)?
R: The correct code for the Δ2bp polymorphism is (-/TG). We corrected the erroneous code in the caption of Table 1.
Line 163: “may putatively originates” should be “may putatively originate”
R: Fixed
Line 271: “in the contest” should be “in the context”
R: Fixed
Line 306: “this isoforms are regulated” should be “these isoforms are regulated” (Plural)
R: Fixed
Line 397: “to what found” should be “to what was found”
R: Fixed
Line 407: “has been implicate” should be “has been implicated”
R: Fixed
Line 443: “pro-inflammatory stimuli down-regulates” should be “pro-inflammatory stimuli down-regulate” (Plural)
R: Fixed
Line 444: “by an NF-kB dependent mechanisms” should be “by an NF-kB dependent mechanism” (Singular)
R: Fixed
Line 581: “one of the therapeutic target” should be “one of the therapeutic targets” (Plural)
R: Fixed